# Access to Orphan Medicinal Products in Bulgaria: An Analysis of the Positive Drug List and Individual Access Schemes

**DOI:** 10.3390/healthcare13182258

**Published:** 2025-09-09

**Authors:** Rumen Stefanov, Ralitsa Raycheva, Kostadin Kostadinov, Georgi Stefanov, Iva Zdravkova-Aneva, Elizabet Dzhambazova, Georgi Iskrov

**Affiliations:** 1Department of Social Medicine and Public Health, Faculty of Public Health, Medical University of Plovdiv, 4002 Plovdiv, Bulgaria; rumen.stefanov@mu-plovdiv.bg (R.S.); kostadinr.kostadinov@mu-plovdiv.bg (K.K.); iva.zdravkova-aneva@mu-plovdiv.bg (I.Z.-A.); elizabet.dzhambazova@mu-plovdiv.bg (E.D.); georgi.iskrov@mu-plovdiv.bg (G.I.); 2Institute for Rare Diseases, 4017 Plovdiv, Bulgaria; g.stefanov@raredis.org; 3Environmental Health Division, Research Institute, Medical University of Plovdiv, 4002 Plovdiv, Bulgaria; 4Department of Social Medicine and Health Management, Faculty of Public Health, Medical University of Pleven, 5800 Pleven, Bulgaria

**Keywords:** orphan designated medicines, rare diseases, regulatory timelines, health technology assessment, national health insurance fund, alternative access

## Abstract

**Background/Objectives**: Orphan medicinal products offer essential treatments for rare diseases, but patient access varies across European Union countries despite a common regulatory framework. In Bulgaria, access is primarily through inclusion in the positive drug list following health technology assessment or via individual access schemes under Ordinance No. 2/2019, which allows for ad hoc reimbursement. This study evaluates the timeliness and extent of Bulgarian patient access to orphan-designated drugs authorized by the European Medicines Agency. **Methods**: We analyzed European Medicines Agency-authorized orphan drugs between July 2006 and September 2023 using data from the European Medicines Agency, Bulgarian health technology assessment bodies, positive drug list records, and individual access scheme reports. Medians, interquartile ranges, stratified analyses, and permutation/bootstrapping methods were applied. **Results:** Of the 142 European Medicines Agency-approved orphan drugs, only 41 (28.9%) were included in the Bulgarian positive drug list. The median time to positive drug list inclusion was 828 days, with pre-health technology assessment delays (median 570 days) as the main bottleneck. Health technology assessment evaluations had a median duration of 204 days. Cancer and accelerated-assessment drugs reached health technology assessment faster, while conditional approvals faced longer delays. Twenty-four drugs were accessed through individual schemes; twenty remained outside the positive drug list. Overall, 43.0% of orphan drugs reached Bulgarian patients via either mechanism. **Conclusions**: Access to orphan drugs in Bulgaria is limited and delayed, mainly due to pre-health technology assessment lags. In light of the forthcoming European Union health technology assessment regulation, Bulgaria must ensure that national processes are capable of rapidly translating centralized assessments into meaningful patient access.

## 1. Introduction

The outcomes of both safe and effective therapy encompass enhancements in health and an extension of lifespan. Consequently, efficient healthcare systems cannot be established without facilitating patient access to lifesaving or life-transforming medicines [1]. Access to innovative treatment for patients is dependent upon several legislative procedures. Securing market authorization (MA) is the initial stage of obtaining healthcare insurance coverage for a product. Subsequently, after the medicine has achieved reimbursement, patients can obtain it through prescriptions and following treatment protocols [2]. All of the collaborating institutions participating in these procedures experience unique challenges in regard to orphan medicines used to treat rare diseases—with a significant number of diseases manifesting a genetic component, and they can be either deadly or very disabling; an estimated 27–36 million people in the European Union (EU) are living with one of these diseases [3]. In 2000, the EU implemented Regulation No. 141/2000 to foster the development of orphan drugs [4]. The United States Orphan Drug Act significantly inspired this initiative, which provides financial incentives to promote the discovery, development, and marketing of medications for rare and severe diseases lacking sufficient treatment options [5]. During the drug development process, the medications are designed as orphan drugs. The next step is to conduct an evaluation in order to acquire a European MA. Between the year 2000 and the present, the European Commission (EC) has already granted approval for more than two hundred orphan drugs which are intended to help patients who are afflicted with rare diseases [6].

Although the EU provides a unified regulatory framework and a standardized approval timeline for orphan medications, individuals from different member states suffering with rare diseases do not have equitable access to orphan drugs [7]. This is due to the possibility of a prolonged wait spanning years for the local country’s post-approval and reimbursement process before they can access the prescription, or they may be unable to obtain it entirely in case the product is not submitted or reimbursed. When the availability of orphan drugs in various European countries is compared, there are significant disparities [8]. In Germany, for instance, 55 of the 61 orphan medications that were authorized by the European Medicines Agency (EMA) between the years 2018 and 2021 were made available to the country, while patients in Lithuania did not have access to any of them. Across Central and Eastern Europe, orphan drugs are not widely available, and the differences are noticeable [9]. Far from the EU Commission’s aim of enhancing access in all member states, the average period for the reimbursement of orphan drugs in the EU is 1.7 years, with a range of as little as three months to as much as 2.5 years [10]. The illustrated disparities in availability and access, indicating that countries such as Germany, the UK, France, Austria, Sweden, and Italy exhibit a significant presence of orphan medicines in their markets, with more than 100 orphan medicines accessible, were in alignment with the findings reported in a prior study conducted by the EC [11]. This pattern was detected in even earlier research conducted in the field: between the years 2005 and 2014, a total of 125 drugs were approved for the treatment of rare diseases, of which 71 were designated as orphans by the EU [12]. Within countries such as Germany, the United Kingdom, Italy, France, and Scandinavian countries, there were between 70 (63%) and 102 (91%) orphan products available. To put that into perspective, only 27–38 percent of these authorized drugs were available in countries such as Greece, Ireland, Bulgaria, Romania, and Croatia [13]. In the latest IQVIA report on EFPIA Patients W.A.I.T. Indicator 2024 Survey, we noticed a positive shift regarding the orphan availability by approval year (2020–2023) in Bulgaria. As observed in the previous report period, in Germany, 59 of the 66 orphan medications that were authorized by the EMA were made available, followed by Italy (n = 50), Austria (n = 46), France (n = 45), and in eight place is Bulgaria, with 37 medicines that account for 56% of the total [14].

It is obvious that the rare disease market access environment is an increasingly prominent agenda item for payers, health technology assessors, health authorities, clinical experts, and all other relevant stakeholders. Beyond regulatory and procedural factors, access to orphan medicines is also shaped by underlying economic challenges. Orphan drugs illustrate features of market failure, where the limited patient population does not provide sufficient incentive for market entry without supportive legislation [15]. Moreover, delays in access entail an opportunity cost, as patients are deprived of timely treatment that could otherwise improve survival and quality of life [16]. For smaller EU health systems such as Bulgaria, the overall budget impact of orphan drugs may be relatively modest in absolute terms, yet the high per-patient cost can exert substantial pressure on constrained healthcare budgets [17]. This imbalance often complicates pricing and reimbursement negotiations, resulting in further delays and inefficiencies in patient access [18].

There are two important aspects to consider when discussing availability. The first is whether the product becomes available at all at any point in time, whereas the second is when the product becomes available (time to market) [11].

In Bulgaria, access to orphan pharmaceuticals is provided by two mechanisms. Since late 2015, Ministerial Order N9 has mandated health technology assessment (HTA) for new medications to be eligible for inclusion in the positive drug list (PDL), which serves as the prerequisite for reimbursement by the National Health Insurance Fund (NHIF). The framework was reinforced in April 2019 with the inclusion of HTA procedures in the Medicinal Products in Human Medicine Act [8]. In 2019, the Ministry of Health promulgated Ordinance No. 2/27 March 2019 (Ordinance), empowering the NHIF to reimburse Bulgarian citizens for medical and other services not covered by compulsory health insurance and not included in Art. 82, para. 1 of the Health Act, for treatment domestically or internationally where no alternative state funding mechanisms exist. Individuals under 18 are eligible for medical care beyond mandatory health insurance, encompassing medical devices, specialized appliances for personal use, dietary foods for specific medical requirements, and pharmaceutical products not enumerated in Article 262, paragraph 1 of the Law on Medicinal Products in Human Medicine. This ordinance mandates NHIF to cover treatments for oncological, hematological, and rare diseases with medicinal products and specialized dietary foods beginning before the age of 18 until completion. All these services are remunerated after NHIF clearance and are in accordance with this ordinance [19,20].

The aim of this study was to evaluate access for Bulgarian patients to orphan drugs by assessing the cut-off for three time intervals—first, from the authorization of EMA orphan-designated medicines to the application for local HTA; second, from HTA application to the final HTA decision; and third, from the HTA decision to the listing on the PDL. Additionally, the study analyzed access obtained directly from the NHIF in accordance with Ordinance No. 2/2019.

## 2. Materials and Methods

### 2.1. Data—Sources, Collection, and Extraction

The analysis utilized data from (1) European Public Assessment Reports (EPAR); (2) HTA dossiers, publicly available on the National Council on Prices and Reimbursement of Medicinal Products (NCPRMP) website (https://ncpr.bg/en/, accessed on 26 March 2024); (3) the PDL; and (4) the NHIF dataset, obtained via official requests, covering 2485 patient-specific approvals for exception-based therapies under Ordinance No. 2 (Ex-Children’s Fund, enacted 2019) from May 2019 to April 2024.

The study focused on orphan-designated (ODD) medicines authorized by the EMA between July 2006 and September 2023. The list of EMA-approved ODDs was cross-referenced with those undergoing HTA in Bulgaria and included in the PDL. Official requests to the NHIF identified ODDs accessed through individual patient schemes under Ordinance No. 2 (enacted 2019), which allows for reimbursement for medicines with active EMA authorization but not yet in the PDL or with initiated HTA via direct negotiations by an NHIF expert committee. Three key time intervals (in days) were defined as follows: (1) time from EMA marketing authorization to HTA dossier submission to NCPRMP; (2) duration from HTA submission to final HTA decision; and (3) time from HTA decision to PDL inclusion.

### 2.2. Statistical Methods

All analyses were conducted using R version 4.5 [21] with the tidyverse package suite [22] employed for data manipulation, statistical modeling, and visualization.

The proportion of EMA-authorized ODDs included in the Bulgarian PDL was calculated, along with cumulative distributions and annual inclusion rates over the study period. The number of patients accessing ODDs through individual patient schemes was quantified, with cross-referencing to identify medicines not subsequently included in the PDL.

Access time from EMA authorization to PDL inclusion was calculated for all ODDs in the PDL. For 36 ODDs with positive HTA decisions, two time intervals were analyzed, namely (1) EMA authorization to HTA initiation and (2) HTA duration, using HTA start and end dates, with the final decision corresponding to PDL inclusion. For 4 ODDs with negative HTA decisions and known durations (onasemnogene abeparvovec, budesonide, darvadstrocel, burosumab), HTA timeliness was calculated. For 5 ODDs included before HTA regulations (eftrenonacog alfa, sorafenib, pasireotide, bedaquiline, tafamidis), only the overall time frame was computed. For budesonide, with an incomplete HTA but a known start date (26 April 2024), the first time point component was calculated. Four ODDs (mosunetuzumab-axgb, mogamulizumab-kpkc, epcoritamab, olipudase alfa) with initiated HTAs but unavailable start dates due to withheld data were excluded from the analysis.

Time intervals were summarized using medians, interquartile ranges (IQRs), and ranges. The proportion of the HTA time interval within the overall time frame was estimated using binomial testing (95% CI). Differences in time interval components were assessed using permutation tests (4000 permutations, two-sided *p*-value). Time intervals were stratified by medicine characteristics (cancer designation, conditional approval, additional monitoring, accelerated assessment). Exceptional circumstance approval was excluded due to limited cases (n = 2: asfotase alfa, tafamidis).

Differences in time intervals were assessed using Wilcoxon rank-sum tests. Additionally, a bootstrap analysis (with 4000 resamples with replacement) was used to estimate median differences in time intervals across stratification factors.

## 3. Results

### 3.1. Time Analysis for PDL

Between July 2006 and September 2023, the EMA authorized 142 medicines with an orphan designation. Of these, 41, representing 28.9%, were included in the Bulgarian PDL, enabling access for patients under the general healthcare scheme. Only three medicines (2.1%) were rejected following the HTA procedure, which were onasemnogene abeparvovec, darvadstrocel, and burosumab. The median time from EMA authorization to inclusion in the PDL was 828 days, roughly equivalent to 2.3 years (Figure 1). The pre-HTA phase had a median duration of 570 days, approximately 1.6 years.

The HTA process, spanning from the HTA submission to the issuance of the final HTA decision, had a median duration of 204 days, around 7 months. The majority of EMA-approved ODDs (n = 98, 69%) were not included in the PDL. Except for the three negative HTA recommendations, we found no public record for the remaining 95 ODDs. We cannot state the precise reason for the lack of access in those cases. However, it is very likely that the MA holder did not submit an HTA dossier for assessment and reimbursement decision-making in Bulgaria.

Differences in the time to access for medicines based on their regulatory characteristics are presented in Figure 2. Medicines designated for cancer had a shorter median time to HTA initiation, at 529 days, compared to non-cancer medicines, which took 651 days. Medicines receiving accelerated assessment from the EMA also experienced a shorter median time to HTA initiation, at 525 days, compared to 619.5 days for those without accelerated assessment. In contrast, medicines with conditional approval faced longer intervals, with a median time to HTA initiation of 691 days and an overall access time of 923.5 days, compared to 529 days and 760 days, respectively, for those without conditional approval (Appendix A Table A1).

### 3.2. Analysis of Access Through Individual Patient Schemes

Beyond the PDL, orphan-designated medicines in Bulgaria can also be accessed through individual patient schemes, as established by Ordinance 2 from 2019 (Appendix A Table A2). These schemes facilitate reimbursement for medicines not included in the PDL, particularly benefiting children with rare diseases. During the study period, 24 orphan-designated medicines were accessed via these schemes. Of these, 20 were available exclusively through individual schemes and had not been added to the PDL by the end of the study period. For four specific medicines—elexacaftor + tezacaftor + ivacaftor, nusinersen, lanadelumab, and ataluren—individual access was granted prior to their PDL inclusion, with a median lead time of 223.5 days, ranging from 58 to 384 days.

### 3.3. Summary of Access Through Both Mechanisms

By the end of the study period, 61 out of the 142 EMA-authorized ODDs, equating to 43.0%, were accessible to Bulgarian patients (Figure 3). This total comprises 41 medicines included in the PDL and 20 available exclusively through individual patient schemes. Additionally, for four ODDs, individual schemes provided access before PDL inclusion, offering patients earlier availability by a median of 223.5 days.

## 4. Discussion

### 4.1. Delays in Access to Orphan Drugs and the Role of Alternative Pathways

Access to orphan drugs varies significantly across countries due to differences in national policies, healthcare infrastructure, and economic resources. More importantly, the processes and criteria for coverage decision-making differ, affecting both the speed and extent to which these therapies become available and accessible across jurisdictions [23,24].

Several key factors contribute to these disparities in access. Economic considerations play a major role, as wealthier nations are able to allocate more resources to healthcare and offer greater incentives to pharmaceutical industry to develop and distribute orphan drugs. Regulatory frameworks also vary, with some countries adopting more streamlined and supportive policies for the approval and reimbursement of ODDs. Additional determinants include the presence of national orphan drug policies, the efficiency of the healthcare system, and the level of government support for rare disease research and treatment. Furthermore, market size and the strategic decisions of pharmaceutical companies regarding where to launch their products also influence overall access to orphan drugs [23,24,25].

We identified 142 ODDs that received market approval from the EMA between July 2006 and September 2023. Although all of these products were authorized via a centralized procedure valid across all EU Member States [26], only 41 (28.9%) are considered routinely accessible to Bulgarian patients by inclusion in the PDL. Given the key differences outlined above, this outcome cannot be directly compared with those of other countries or regions. Nevertheless, access analyses and jurisdiction-level comparisons could help identify tailored policies and tools to enhance access to orphan drugs both globally and locally [23,24].

Our findings suggest that the primary contributor to delays in patient access to orphan medicinal products in Bulgaria is the prolonged interval preceding the initiation of health technology assessment. Specifically, the median duration from the EMA marketing authorization to HTA submission was 570 days, while the duration of the HTA evaluation itself (median 204 days) did not represent a significant obstacle within the overall process. Therefore, the Bulgarian HTA evaluation process alone is unlikely to constitute the critical bottleneck. Rather, the substantial delay between EMA authorization and HTA initiation can mainly be attributed to delayed or absent submissions of HTA dossiers by pharmaceutical companies [27]. In smaller markets such as Bulgaria, characterized by limited healthcare budgets and relatively low potential commercial returns, pharmaceutical companies may lack adequate incentives to promptly initiate the local HTA process and subsequent inclusion in the PDL [28].

Comparable challenges have been addressed in other EU health systems by coupling early or conditional access with real-world evidence (RWE) generation that feeds back into reimbursement decisions. In England, nusinersen entered under a managed access agreement with mandated data collection. A subsequent review of accumulated evidence led NICE to broaden clinical eligibility, illustrating how RWE can reshape access while formal listing decisions mature [29]. In Scotland, the ultra-orphan pathway permits up to three years of access while further effectiveness data are gathered, followed by reassessment [30]. Italy’s AIFA registries operationalize outcome tracking to support managed entry agreements (including for ODDs) and inform re-evaluation [31]. France’s early access program (formerly ATU, now AAP) similarly ties early publicly funded access to post-listing evidence obligations and HAS reassessment [32]. Spain’s VALTERMED platform demonstrates national infrastructure for outcome-based agreements, even as its role in price/reimbursement decisions continues to evolve [33]. Together, these models show that structured RWE can de-risk uncertainty without deferring patient access.

In Bulgaria, a similar mechanism exists. The individual patient access schemes regulated by Ordinance No. 2 from 2019 have emerged as crucial alternative pathways, providing expedited patient access to orphan therapies [20]. These schemes effectively mitigate the negative impact of delayed or absent PDL inclusion, offering a more flexible mechanism to deliver urgently required, often lifesaving treatments. Furthermore, the interval during which patients gain access to orphan therapies under Ordinance 2 facilitates the accumulation of valuable real-world clinical experience, including data on drug effectiveness, safety profiles, and patient outcomes [34]. The utility of such evidence is clearly illustrated by the subsequent inclusion in the PDL of four therapies initially accessible exclusively via these individual schemes.

Given these findings, it is important for policymakers, particularly in EU member states with a lower gross domestic product (GDP) and constrained healthcare budgets, to consider whether alternative, more flexible access mechanisms such as individual patient schemes could potentially offer greater efficiency compared to conventional HTA-driven pathways [35]. Expanding the use of such flexible solutions could not only enhance timely patient access but also enable systematic collection of real-world clinical data. Therefore, policy development efforts should focus on creating tailored regulatory frameworks that effectively integrate traditional HTA approaches with responsive individual patient schemes, optimizing resource allocation, equity in patient access, and the sustainability of orphan drug reimbursement.

These considerations also align with the forthcoming implementation of the EU HTA Regulation (EU) 2021/2282 [36]. While joint clinical assessments (JCAs) will harmonize the evaluation of clinical evidence at the European level, they will not eliminate the national responsibilities of pricing, reimbursement, and dossier submission [37]. In smaller markets such as Bulgaria, delayed or absent HTA submissions may therefore persist unless incentives and procedures are adapted to ensure timely uptake of JCA outputs. Embedding flexible mechanisms like Ordinance No. 2 within this evolving framework could help bridge the gap between centralized evidence generation and local reimbursement decisions, ensuring that the regulation translates into real improvements in patient access.

### 4.2. The Rationale for Orphan Designation

Orphan designation is a critical regulatory mechanism designed to encourage the development of drugs for rare diseases [38,39,40,41]. Under conventional market conditions, these diseases often lack sufficient incentives for pharmaceutical companies to invest in research and development. By granting orphan designation, health authorities provide benefits that help offset the high costs and risks associated with developing treatments for small patient populations [38,41,42,43]. Since the adoption of the Orphan Medicinal Products Regulation, the EU has witnessed significant progress in the development and availability of orphan drugs. However, the regulation has also faced challenges, including insufficient focus on areas with the greatest unmet medical need and rising healthcare costs [44].

One recurring criticism is that these incentives contribute to the subsequent high prices of orphan drugs, rendering treatments unaffordable for patients with rare diseases and challenging healthcare systems [42,43,45]. Additionally, some argue that the criteria for orphan designation can be exploited by pharmaceutical companies to gain market exclusivity for drugs that may not genuinely benefit rare disease populations [42,43,46]. Concerns have also been raised regarding the efficiency and transparency of the designation process, with calls for more rigorous scrutiny and clearer guidelines [38,46,47,48].

Given these concerns, it is anticipated that the criteria and processes for orphan designation will evolve. Recent evaluations have emphasized the need for updates that more effectively target areas of unmet medical need and incorporate scientific and technological advances [44]. Potential changes may include more stringent requirements for demonstrating the rarity and medical necessity of the condition being treated, as well as stronger safeguards against the misuse of orphan designation for commercial advantage [38,39,40,41,42,43,46,48]. These updates aim to refine the orphan designation process and ensure it continues to fulfill its original goal of supporting patients with rare diseases.

### 4.3. The Role of HTA

HTA plays a critical role in determining the accessibility of orphan drugs [49]. HTA bodies evaluate the clinical effectiveness, cost-effectiveness, and broader impact of new treatments on the healthcare system. For orphan drugs, which are often associated with high costs and evidence uncertainty, HTAs are essential for justifying their value and securing reimbursement from healthcare payers [50]. The process involves a rigorous analysis of available data, including clinical trial results and real-world evidence, to ensure that the benefits of a drug outweigh its costs. This assessment supports efficient resource allocation and enables informed decisions regarding which orphan drugs should be funded [49,50,51,52].

The HTA process, along with its methods and criteria, has been cited as a contributing factor to disparities in access to orphan drugs [51,52,53]. In 2021, the European Parliament adopted a new EU HTA regulation aimed at harmonizing HTA practices across member states by introducing joint clinical assessments (JCAs) and joint scientific consultations (JSCs) [54,55]. This regulation is expected to streamline the evaluation process for innovative health technologies by reducing the redundant efforts and accelerating access to these treatments across the EU [56,57]. Through centralized assessments, the regulation seeks to promote consistent and transparent decision-making, thereby helping to mitigate disparities in access among countries. However, the success of this regulation will depend on effective implementation, sustained cooperation among member states, and the ability to address specific national healthcare needs [56,57].

One major concern is the potential for delays in the assessment process due to the need for coordination across multiple countries. The regulation may encounter difficulties in reconciling the diverse healthcare priorities and economic conditions of different member states [58,59], which could impact the uniformity of orphan drug access. There is also a risk that a centralized assessment process may not fully consider the unique needs of rare disease patients in specific regions, potentially leading to inequities in access [60]. Furthermore, stringent evidence requirements could present barriers for certain orphan drugs, limiting their availability to patients who need them [59,60].

### 4.4. Policies and Tools to Improve Orphan Drugs Access

Effective policies at both the national and supranational levels are essential for improving access to orphan drugs. Innovative pricing tools, along with integrated approaches to treatment and care for rare disease patients, can address this complex issue in a synergistic manner [23,24,25]. One promising strategy is the implementation of value-based pricing models, in which a drug’s price is tied different value elements, including its demonstrated clinical benefits [61,62]. This approach could potentially ensure that patients receive effective treatments without placing excessive financial strain on healthcare systems. In addition, special reimbursement schemes, such as dedicated funds or specific insurance coverage for orphan drugs, can help reduce cost barriers and improve patient access [23,24,25,63]. Engaging communities in pricing decisions may also promote transparency and fairness throughout the process [64].

Exploring alternative access schemes is another way to provide sustainable financial support for the development and accessibility of orphan drugs [65]. Collaborations among governments, pharmaceutical companies, and nonprofit organizations can leverage combined resources and expertise to accelerate orphan drug development [23,24]. Moreover, outcome-based coverage, where payments are contingent on achieving specific health outcomes, can encourage the creation of effective therapies while ensuring the efficient use of public and private funds [66,67]. These models can help address the financial hurdles associated with orphan drug development and improve patient access to necessary treatments.

Finally, fostering collaborative networks and integrated care models can greatly enhance access to orphan drugs. Centers of expertise for rare diseases, which unite multidisciplinary teams of healthcare professionals, researchers, and patient advocacy groups, can improve the diagnosis, treatment, and long-term management of rare diseases [68,69,70]. These networks enable the exchange of knowledge, resources, and best practices, contributing to improved patient outcomes [71,72]. Integrated care models featuring coordinated services across healthcare sectors can ensure that patients receive holistic, continuous care, ultimately enhancing their overall quality of life [73].

Our findings suggest two specific points of the ODD access process in Bulgaria that need further research. First, the delayed or missing HTA submissions at the local level. EU and national authorities should engage in dialog with MA holders to identify specific policy measures to address this problem. Second, the efficient application of the individual access scheme. Reimbursement decision-makers and pharmaceutical companies should explore means to better use the RWE generated through this mechanism.

### 4.5. Limitations of the Study

This study has several limitations that should be acknowledged. First, the analysis was limited to publicly available data from national and European regulatory bodies, which may be subject to reporting longer time intervals or incomplete documentation. For several ODDs, HTA initiation dates or final decisions were not disclosed, leading to the exclusion of some products from the full time frame analysis. This could introduce selection bias and may underestimate or overestimate the true extent of access time intervals.

Second, we did not explore in more detail the ODDs that were found missing from the Bulgarian PDL. There was no public record in the case of 95 EMA-approved ODDs that were not included in this list. The rationale behind this situation is unclear and needs further research. However, it is very likely that the MA holder did not submit an HTA dossier for assessment and reimbursement decision-making in Bulgaria.

Third, while the analysis assessed time intervals and stratified them by key regulatory characteristics, it did not incorporate clinical or budget impact data, which often influences reimbursement decisions. Additionally, patient-level outcomes and real-world utilization were beyond the scope of this study but are essential for evaluating the true impact of access on public health.

Lastly, the findings are context-specific and reflect the regulatory and reimbursement environment in Bulgaria. While some insights may be generalized to other EU countries, differences in healthcare system structures, HTA practices, and funding policies limit direct cross-country comparisons.

## 5. Conclusions

This study provides the first comprehensive analysis of access timelines for EMA-authorized ODDs in Bulgaria, highlighting substantial delays and limited availability under the national reimbursement framework. Despite a harmonized EU regulatory pathway, only 28.9% of approved ODDs were included in Bulgaria’s PDL over the study period. The median time from European authorization to national reimbursement exceeded two years, with the most pronounced delay occurring prior to the initiation of the HTA process. These findings underscore a critical bottleneck in transitioning from EMA approval to local evaluation and reimbursement.

Regulatory characteristics such as cancer indication and accelerated assessment status were associated with shorter access timelines, while conditionally approved products experienced prolonged delays. In parallel, individual patient access schemes under Ordinance No. 2 facilitated earlier availability for selected therapies, particularly in pediatric populations. However, these schemes often served as temporary or alternative routes, with most ODDs accessed through them remaining outside the PDL by the end of the study period.

The results reflect systemic challenges within Bulgaria’s regulatory and reimbursement processes for rare disease therapies. To improve patient access, targeted measures are needed to reduce pre-HTA delays, enhance procedural efficiency, and prioritize therapies addressing high unmet medical needs. As the EU moves toward joint clinical assessments under the new HTA regulation, national systems must be prepared to integrate centralized evaluations into streamlined and timely reimbursement decisions. Strengthening the alignment between regulatory and funding pathways is essential to ensure equitable, sustainable, and prompt access to orphan drugs for patients with rare diseases in Bulgaria.

## Figures and Tables

**Figure 1 healthcare-13-02258-f001:**
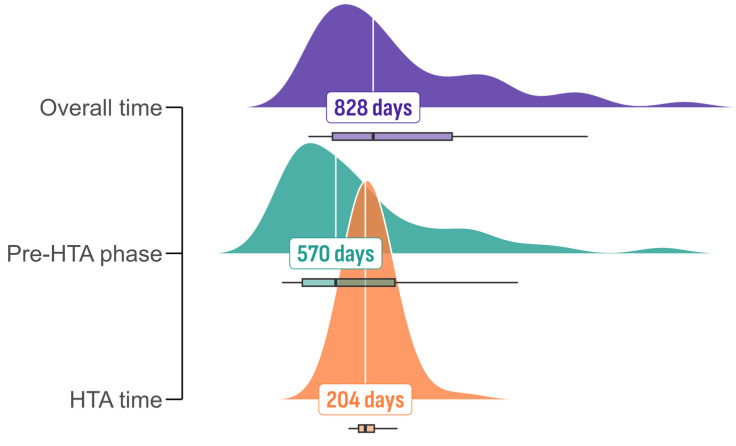
Distribution of time frame between EMA marketing authorization and inclusion in the Bulgarian PDL.

**Figure 2 healthcare-13-02258-f002:**
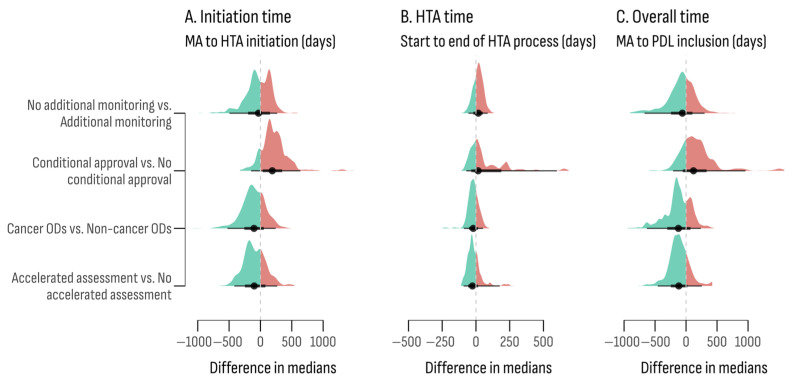
Differences in medians of time point estimations according to additional monitoring, conditional approval, and accelerated assessment status.

**Figure 3 healthcare-13-02258-f003:**
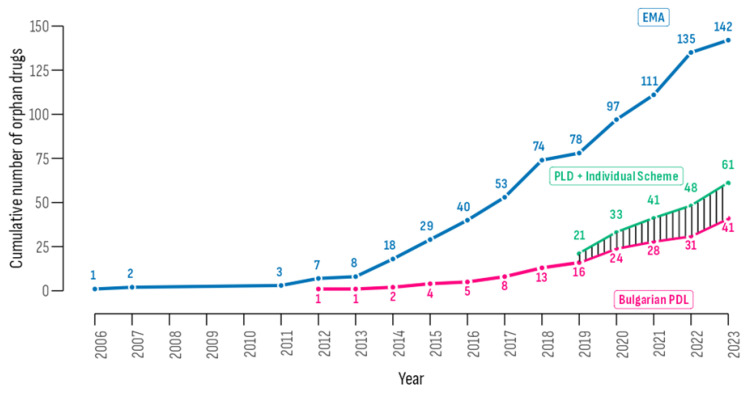
Comparison of the cumulative number of orphan drugs authorized by EMA and included in the Bulgarian PDL or individual access scheme.

## Data Availability

Research data can be obtained upon request.

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
