# Peer review of "Access to Orphan Medicinal Products in Bulgaria: An Analysis of the Positive Drug List and Individual Access Schemes"

_healthcare, 2025, doi:10.3390/healthcare13182258_

Round 1

Reviewer 1 Report

Comments and Suggestions for Authors

the manuscript is interesting but following comments need to be addressed before publication;

Is Orphan Medicinal Products an organization?

What are orphan drugs?

What is the meaning of the main bottleneck here?

Please provide which software packages were used for data analysis. And what was their function that was used for analysis?

Statistical analysis part is too long.

What is reimbursement for medicines?

What is the future perspective of the study?

Please provide the significance of the study.

Author Response

We sincerely thank the reviewer for the valuable and constructive comments. The feedback has greatly helped us to improve the clarity, accuracy, and overall quality of the manuscript. We have carefully considered all suggestions and revised the text accordingly. We believe that these changes have strengthened the work and hope that the revised version addresses the reviewers’ concerns satisfactorily. Please find all your comments and suggestions addressed in the uploaded word file.

Reviewer 2 Report

Comments and Suggestions for Authors

To further strengthen the manuscript and address the points above, I offer the following constructive suggestions:

Clarify HTA Submission Outcomes: Enrich the results (or discussion) by explicitly stating how many of the EMA-authorized orphan drugs were submitted for HTA in Bulgaria during the study period, and of those, how many received positive versus negative decisions. This context will highlight that the main barrier is the lack of submission. For instance, if data permit, the authors could say: “Out of 142 OMPs, X were submitted for HTA review; Y of these achieved a positive HTA recommendation and PDL inclusion (41 are listed), while Z received negative decisions or were withdrawn.” Even a qualitative statement that “virtually all drugs that underwent HTA were eventually listed, indicating that not applying at all was the dominant reason for non-access” would support the paper’s conclusions about pre-HTA delays.

Emphasize the Extent of Unmet Need: The authors might consider highlighting more explicitly that over half of orphan medicines (57%) had no reimbursed access in Bulgaria by 2023. While the 43% accessible figure is given, flipping it around underscores the gap in a more striking way. This could be mentioned in the discussion or conclusions to reinforce the public health importance of the issue – that a majority of rare disease therapies authorized at the EU level never reached Bulgarian patients through any funding mechanism. If any of these non-accessible drugs address severe conditions, it might be worth illustrating the human impact (without going off scope): e.g., “This means patients with dozens of rare conditions likely had no locally funded treatment option.” Such context can drive home the urgency.

Discuss Time Trends or Recent Developments: If data allow, comment on whether access is improving in recent years. The Introduction alludes to a positive trend (56% availability for 2020–2023 approvals). The authors could integrate this by noting in the discussion that their long-term average of 28.8% masks an upward trend in the later years, possibly due to the establishment of HTA processes and Ordinance 2. For example: “Notably, there are signs of improvement: of the orphan drugs authorized in the last 3–4 years of our study, a higher proportion have already gained access in Bulgaria (as high as ~50% according to recent reports). This suggests recent policy efforts and the Ordinance 2 scheme may be yielding some progress, though substantial gaps remain.” This addition would acknowledge progress and position the study as a baseline on which future data can build.

Streamline and Focus the Discussion: Consider narrowing the scope of the discussion to maintain focus on actionable insights from the study. Some sections could be condensed. For example, Section 4.2 (on orphan designation) could be shortened or partially moved to the introduction as background, since it does not directly use the study’s findings. The part of Section 4.4 on integrated care networks, while insightful, might be beyond the scope of access timelines; it could be trimmed to briefly note that broader healthcare system strengthening (centers of expertise, etc.) complements access policies. Refocusing these sections will ensure the reader’s attention stays on the core findings about access delays and solutions for Bulgaria.

Policy Recommendations: Building on the findings, the authors might strengthen their policy recommendation section (end of discussion) with a bit more specificity. For instance, since delayed HTA submissions are the key problem, suggest mechanisms to incentivize or require earlier submissions in small markets. This could include: fast-track negotiation frameworks for orphan drugs, regional collaborations where several smaller countries pool resources to assess and negotiate access, or financial incentives/penalties for marketing authorization holders to file for reimbursement within a certain timeframe after EMA approval. The manuscript already mentions flexible pathways and integrating individual schemes with HTA. Adding a sentence or two about encouraging manufacturer engagement (perhaps referencing the upcoming joint EU HTA as a chance to reduce duplicative effort for companies) would directly address the root cause identified. Similarly, since the individual funding scheme is heavily utilized, one could recommend making its outcomes more systematically feed into PDL decisions (as seen with four drugs that eventually transitioned). In short, a more pointed set of suggestions for Bulgarian policymakers – e.g., “establish a clear mandate or incentive for timely HTA submission, possibly by leveraging the new EU joint clinical assessments to simplify the process for companies” – would enhance the impact of the paper’s conclusion.

Author Response

We sincerely thank the reviewer for the valuable and constructive comments. The feedback has greatly helped us to improve the clarity, accuracy, and overall quality of the manuscript. We have carefully considered all suggestions and revised the text accordingly. We believe that these changes have strengthened the work and hope that the revised version addresses the reviewers’ concerns satisfactorily. Please find all your comments and suggestions addressed in the uploaded Word file.

Reviewer 3 Report

Comments and Suggestions for Authors

Thank you for submitting your manuscript to the journal. Overall, it is well-prepared and clearly written. However, I have a few suggestions that may help enhance and strengthen the quality of the paper:

Abstract
The abstract is well-crafted, and the keywords are appropriate, though the number of keywords might be slightly excessive.

Introduction
The introduction is acceptable and appears to incorporate a relevant literature review.

  • Several general statements within the introduction would benefit from appropriate citations to substantiate their validity (e.g., lines 41-43; 66-67). Check figures numbering as well.
  • To add greater depth, consider introducing basic economic concepts early on, such as market failure, the opportunity cost of delayed access, and the pricing and reimbursement challenges faced by smaller EU economies. For example, it could be noted that in smaller health systems, the limited overall budget impact of orphan drugs may be overshadowed by their high per-patient costs, resulting in access delays and inefficiencies over time.
  • While the study’s aims are stated, the introduction would benefit from a clearer emphasis on why Bulgaria is a particularly important case study—highlighting its underrepresentation in orphan drug access research and its distinctive dual-pathway system. It would also be helpful to explicitly state what this study adds to the existing EU orphan drug policy literature, such as the granular time-based analysis and examination of ad hoc access schemes.
  • Consider including a brief paragraph at the end of the introduction or literature review section that outlines the structure of the manuscript, providing a roadmap for readers.

Materials and Methods
This section is generally satisfactory.

Results
The results are clearly presented and appropriately based on descriptive analysis.

Discussion

  • Section 4.1 would benefit from a more thorough discussion. The first paragraph, in particular, may be unnecessary. It is important to frame your findings within real-world contexts—without this, policy suggestions risk seeming unsupported. Incorporating examples or references to studies where real-world evidence has influenced Positive Drug List (PDL) inclusion in other settings could strengthen this section.
  • Additionally, explicitly linking your findings to the current or proposed implementation challenges of EU Health Technology Assessment (HTA) regulations in smaller markets would enhance relevance.
  • Sections 4.2 to 4.4 might be better placed earlier in the manuscript, either as part of the introduction or as a separate literature review section, since they diverge somewhat from the core discussion of results and applications.
  • Section 4.5, which covers the study’s limitations, would be more appropriately placed at the end of the conclusions section.

References
The reference list could be improved by incorporating more recent and diverse sources.

Author Response

(The authors gave the same response as above.)
